# Examining the Drivers of Racial/Ethnic Disparities in Non-Adherence to Antihypertensive Medications and Mortality Due to Heart Disease and Stroke: A County-Level Analysis

**DOI:** 10.3390/ijerph182312702

**Published:** 2021-12-02

**Authors:** Macarius M. Donneyong, Michael A. Fischer, Michael A. Langston, Joshua J. Joseph, Paul D. Juarez, Ping Zhang, David M. Kline

**Affiliations:** 1College of Pharmacy, The Ohio State University, Columbus, OH 43210, USA; 2General Internal Medicine at Boston Medical Center, Boston University School of Medicine, Boston, MA 02118, USA; Michael.Fischer@bmc.org; 3Department of Electrical Engineering and Computer Science, University of Tennessee, Knoxville, TN 37996, USA; langston@tennessee.edu; 4College of Medicine, The Ohio State University Wexner Medical Center, Columbus, OH 43210, USA; Joshua.Joseph@osumc.edu; 5Department of Family and Community Medicine, Meharry Medical College, Nashville, TN 37208, USA; pjuarez@mmc.edu; 6Division of Biomedical Informatics, College of Medicine, The Ohio State University, Columbus, OH 43210, USA; Ping.Zhang@osumc.edu; 7Department of Biostatistics and Data Science, Division of Public Health Sciences, Wake Forest School of Medicine, Winston-Salem, NC 27101, USA; dkline@wakehealth.edu

**Keywords:** adherence, racial disparity, antihypertensives, social determinants of health, determinants of health, hypertension, heart disease, stroke

## Abstract

**Background:** Prior research has identified disparities in anti-hypertensive medication (AHM) non-adherence between Black/African Americans (BAAs) and non-Hispanic Whites (nHWs) but the role of determinants of health in these gaps is unclear. Non-adherence to AHM may be associated with increased mortality (due to heart disease and stroke) and the extent to which such associations are modified by contextual determinants of health may inform future interventions. **Methods:** We linked the Centers for Disease Control and Prevention (CDC) Atlas of Heart Disease and Stroke (2014–2016) and the 2016 County Health Ranking (CHR) dataset to investigate the associations between AHM non-adherence, mortality, and determinants of health. A proportion of days covered (PDC) with AHM < 80%, was considered as non-adherence. We computed the prevalence rate ratio (PRR)—the ratio of the prevalence among BAAs to that among nHWs—as an index of BAA–nHW disparity. Hierarchical linear models (HLM) were used to assess the role of four pre-defined determinants of health domains—health behaviors, clinical care, social and economic and physical environment—as contributors to BAA–nHW disparities in AHM non-adherence. A Bayesian paradigm framework was used to quantify the associations between AHM non-adherence and mortality (heart disease and stroke) and to assess whether the determinants of health factors moderated these associations. **Results:** Overall, BAAs were significantly more likely to be non-adherent: PRR = 1.37, 95% Confidence Interval (CI):1.36, 1.37. The four county-level constructs of determinants of health accounted for 24% of the BAA-nHW variation in AHM non-adherence. The clinical care (β = −0.21, *p* < 0.001) and social and economic (β = −0.11, *p* < 0.01) domains were significantly inversely associated with the observed BAA–nHW disparity. AHM non-adherence was associated with both heart disease and stroke mortality among both BAAs and nHWs. We observed that the determinants of health, specifically clinical care and physical environment domains, moderated the effects of AHM non-adherence on heart disease mortality among BAAs but not among nHWs. For the AHM non-adherence-stroke mortality association, the determinants of health did not moderate this association among BAAs; the social and economic domain did moderate this association among nHWs. **Conclusions:** The socioeconomic, clinical care and physical environmental attributes of the places that patients live are significant contributors to BAA–nHW disparities in AHM non-adherence and mortality due to heart diseases and stroke.

## 1. Background

Blacks/African Americans (BAAs) bear a larger brunt of the burden of hypertension in the US. About 41% to 57% of BAAs are estimated to be living with hypertension while 67–90% are less likely than non-Hispanic Whites (nHWs) to have their blood pressure (BP) under control [1,2]. The recent Heart Disease and Stroke Statistics report (2016) showed that high BP was associated with higher risk of fatal strokes (1.8 times), fatal heart diseases (1.5 times) and end-stage renal disease (4.2 times) among BAAs compared to nHWs [3]. About $46 billion is spent annually on medications and other healthcare services to control high BP in the US. It is therefore imperative to examine the sources of these BAA–nHW disparities in these hypertension-induced outcomes.

Non-adherence to AHM treatment is the major determinant of high BP control and deaths due to heart diseases and stroke among hypertension patients [4,5,6]. BAAs are 43% to 47% less likely to adhere to AHM treatment compared to nHWs [7,8,9]. With a non-adherence rate as high as 47% to 73% among BAAs [7,8,9,10,11,12], it is not surprising that high BP control and adverse hypertension outcomes are poorer among BAAs [13,14,15,16]. However, there are insufficient data on the contributors to BAA–nHW disparities in AHM non-adherence. Further, the extent to which disparities in non-adherence to AHM account for BAA–nHW disparities in hypertension sequelae such as heart disease and stroke mortality is not well understood. Knowledge of the factors that moderate the adverse effects of AHM non-adherence on hypertension sequelae is critical for progress towards eliminating BAA-nHW disparities, yet such data are lacking in published research. 

Determinants of health [17]—a complex range of personal, social, economic, and environmental factors—have been found to be associated with cardiovascular outcomes [18,19,20,21,22,23,24,25,26]. Emerging data have identified county and neighborhood-level determinants of health including poverty, food insecurity, lack of social support, low social affluence, residential instability, socioeconomic disadvantage, and high crime, as potential predictors of nonadherence to AHMs [7,8,9]. Given that these place-based determinants of health are disproportionally higher among BAA communities, it is plausible that these factors may moderate the impact of AHM non-adherence on BAA–nHW disparities in heart disease and stroke mortality [27,28]. This hypothesis has not been tested empirically. 

Therefore, the primary objectives of this study were to: (1) assess the role of the determinants of health on BAA–nHW disparities in AHM non-adherence; (2) quantify the associations between AHM non-adherence and mortality due to heart disease and stroke and assess whether the determinants of health modifies this association between BAAs and nHWs. 

## 2. Methods

### 2.1. Data Sources

The CDC Atlas of Heart Disease and Stroke (2014–2016 cycle) and the 2016 Community Health Rankings (CHR) datasets were linked by unique, five-digit, Federal Information Processing Standard (FIPS) codes of the counties present in both datasets. The CDC Atlas consists of county-level estimates of all heart diseases, mortality, and hospitalizations based on data from the Deaths National Vital Statistics System and Centers for Medicare and Medicaid Services Medicare Provider Analysis and Review (MEDPAR) file, Part A, respectively [29]. This database also contains county-level measurements of risk factors, social and economic factors, health care delivery and insurance and health care costs data derived from multiple data sources [30]. The CHR database is the most comprehensive dataset created specifically to characterize counties by four major domains of determinants of health—health behaviors, clinical care, social and economic factors, and physical environment. 

### 2.2. Measurement of Determinants of Health Factors

The CHR model characterizes communities with respect to how healthy they are (health outcomes) and existing modifiable factors (health factors) that predict future health. For the purposes of ranking counties by these factors, the CHR calculated weighted composite scores for the domains of both health outcomes and community-level determinants of health [30]. The rationale and methods for creating these domains and composite scores have already been published [31]. Briefly, these domains of determinants of health that have been pre-defined by CHR include: (1) health behaviors—factors that improve health (eating well and being physically active) as well as those that increase the risk of diseases (smoking, excessive alcohol intake, and risky sexual behavior); (2) clinical care—defined as access to affordable, quality, and timely health care; (3) social and economic factors—measures of income, education, employment, community safety, and social supports; and (4) physical environment—this domain characterizes the physical environments where people live, learn, work and play, based on the quality of air they breathe, water they drink, houses they live in, and the transportation they access to travel to work and school.

### 2.3. Measurement of County-Level Non-Adherence to AHM

The CDC Atlas dataset includes measures of the proportion of days covered (PDC) with blood pressure medication for a period of 365 days for each county. PDC is a validated measure of adherence and persistence to medications, especially among patients with repeated fills [32,33,34]. The PDC measures in the 2014–2016 CDC Atlas dataset were derived from Medicare Advantage and Medicare fee-for-service beneficiaries aged ≥65 years old who had Medicare Part D coverage in 2015–2016. Further details for the inclusion of Medicare data for the calculation of non-adherence are available at https://www.cdc.gov/dhdsp/maps/atlas/index.htm (accessed on 29 November 2021) [29]. Based on the average county-level PDC measures defined during the 2015–2016 period, we operationally defined the prevalence of county-level AHM non-adherence as PDC < 80% [32,33,34]. 

### 2.4. Measurement of Outcomes

The primary outcome for objective 1 was BAA–nHW disparities in AHM non-adherence, whereas heart disease and stroke mortality were assessed as the primary outcomes in objective 2. 

Quantification of racial disparities in AHM non-adherence (Objective 1): We used the prevalence rate ratio (PRR), a widely used measure of racial disparities, to define the BAA–nHW disparity in AHM non-adherence [35]. For each county that presented measures of prevalence of AHM non-adherence for both BAAs and nHWs, the prevalence among BAAs was divided by that among nHWs to generate county-level PRRs.Assessment of heart disease and stroke mortality (Objective 2): In the CDC Atlas dataset, heart disease mortality was defined as deaths due to diseases of the circulatory system (ICD-10 codes: I00-I99, I11, I13, I20-I51). All deaths for which stroke was identified as the underlying cause were defined as stroke mortality.

Potential confounders: We leveraged county-level demographic factors—percentage ≥65 years old, percentage female and percentage rural—measured in the 2016 CHR database as potential confounders of the associations tested in the analysis for both objectives 1 and 2. 

### 2.5. Statistical Analysis

Conceptual framework (Figure 1): AHM non-adherence is a complex health behavior that is influenced by multiple factors, including health behaviors, social and economic factors, access and quality of care as well as the broader physical environmental context in which patients live. In the conceptual framework depicted in Figure 1A, we posit that determinants of health are associated with BAA–nHW disparities in AHM non-adherence independently of the demographic (BAA population, female population and rural area) make-up of counties. In Figure 1B, we hypothesize that non-adherence to AHM is a predictor of both heart disease and stroke mortality independent of the effects of county demographics and contextual determinants of health. Further, we posit that the AHM non-adherence–mortality relationship is moderated by contextual county-level determinants of health. Therefore, we treated place-based determinants of health both as potential confounders and as potential moderators of the associations between AHM non-adherence and mortality due to heart disease and stroke. To test these hypotheses empirically, we examined 875 counties across 38 states that featured sufficient populations of BAAs to allow for the feasible measurement of heart disease and stroke mortality rates. 

For objective 1, we used a series of hierarchical linear mixed regression (HLM) models to assess the independent associations between constructs of determinants of health and BAA–nHW disparities in AHM non-adherence. We modeled PRR as the dependent variable in the HLM where state ID (representing unique states without state-level factors) was modeled as a random effect (level 1) and county-level constructs of determinants of health as fixed effects (level 2). We built four HLM models to measure the associations between each construct of determinants of health and BAA–nHW disparity in AHM non-adherence in a series of four HLM models that: (1) were unadjusted (Model 1); adjusted for the confounding effects of age (Model 2); adjusted for other determinants of health (Model 3); and adjusted for all constructs of determinants of health as well as the potential confounding effects of age, sex and rural status (Model 4). A pseudo-R^2^ [36] was calculated from the variance components of Model 3 (containing only constructs of determinants of health) to quantify the total variation in BAA–nHW disparities that is explained by state-level random effects and county-level constructs of determinants of health. We implemented these HLM models in STATA version 14 (StataCorp, College Station, TX, USA).

For objective 2, we jointly modeled the log of the rates of mortality per 100,000 for heart diseases and stroke for BAAs and nHWs. We fit a shared component or factor model to the log rates of heart disease and stroke mortality for BAAs and nHWs in each county [37,38,39]. These models were fitted within the Bayesian paradigm using a Markov chain Monte Carlo algorithm and implemented using nimble in R [40]. We computed the overall means and standard deviations of each covariate and used these distributions to standardize each covariate prior to modeling in order to have mean 0 and standard deviation 1. Thus, the coefficient estimates measured from models represent one standard deviation change in each covariate. We considered four race-specific mortality outcomes (in log rate): heart disease mortality for BAAs, stroke mortality for BAAs, heart disease mortality for nHWs, and stroke mortality for nHWs. First, we quantified the associations between AHM non-adherence and each outcome, heart disease and stroke mortality, through a main effects model in which we included determinants of health and demographic factors as potential confounders. Second, we tested whether the determinants of health moderated the association between AHM non-adherence and outcomes by including interaction terms between AHM non-adherence and each health determinant factor (interaction effects model). Finally, we addressed potential unmeasured effects, clustering by area and racial/ethnic groups in all the models, as follows:Because not all measures of areal risk are easily measured, there are likely to be unmeasured factors that contribute to the risk of mortality from heart disease and stroke. These unmeasured factors may also differ by race/ethnicity, as racial groups may experience a common areal environment differently. To capture this variation, we included a county-level, shared racial factor in all models which accounts for correlation across outcomes within a racial group due to unmeasured factors.To account for correlations between races within a county, we allowed the BAA and nHW unmeasured factors within a county to be correlated.In addition, counties within a state were correlated because they share several health determinants, whereas states may differ with respect to these factors. To account for this, we included a state factor that is shared across all the outcomes in a state.

For each factor, we also included factor loadings so that the factor is appropriately scaled for each outcome and constrained particular loadings to be 1 for identifiability. In addition to providing insights about shared latent risk, this model properly accounted for the hierarchical nature of the data. 

Measures of interest: For the main effects model described above, we computed the posterior mean, 95% credible interval (CI) and posterior probability that the effect of a given covariates would be positive. For the interaction effects, the estimated effects (the posterior mean, 95% CI and posterior probability) of each covariate on the effect of non-adherence (i.e., the interaction term) were recorded. Each estimated effect was interpreted as the change in the log rate given the mean values of the other covariates. The posterior mean estimates of the unmeasured factors for each county and race group, and the posterior probability that they are above average (i.e., greater than 0) were computed and mapped.

## 3. Results

Our analysis included 875 counties in the United States that included sufficient population sizes for their rates of heart disease and stroke to be released for both BAAs and nHWs. The distribution of the mean and standard deviations of the county-level prevalence of AHM non-adherence, determinants of health, and demographic factors are presented for all counties and by regions in Table 1. Counties in the South exhibited higher means of health behavior, clinical care, and social and economic factors scores. The counties in the South also were more rural and featured higher proportions of BAAs than other regions. The distribution of the individual features that make up these constructs of determinants of health are presented in Appendix A. The overall PRR was 1.37, 95% Confidence Interval (CI):1.36, 1.37 (Appendix A). There was a regional variation in BAA-nHW disparities—the largest disparities were observed in the Midwest (PRR = 1.50, 95% CI: 1.48, 1.53) and East (PRR = 1.49, 95% CI: 1.47, 1.52) as compared to the South (PRR = 1.33, 95% CI: 1.33, 1.34) and West (PRR = 1.33, 95% CI: 1.30; 1.37), Appendix A. 

### 3.1. The Role of County-Level Constructs of Determinants of Health in BAA–nHW Disparities in AHM Non-Adherence

Collectively, the four county-level constructs of determinants of health accounted for 24% of the BAA–nHW variation in AHM non-adherence (Table 2). While all but the physical environment construct were significantly associated with BAA–nHW disparities in AHM non-adherence in the bivariate (Model 1) and age-adjusted models (Model 2), only clinical care (β = −0.21, *p* < 0.001) and social and economic constructs (β = −0.11, *p* < 0.01) were inversely associated with BAA–nHW disparities in AHM non-adherence after controlling for other determinants of health and the potential confounding effects of age ≥65, gender, and rural location (Model 4), Table 2. This means that the BAA–nHW disparities decreased with higher prevalence of better clinical care and social and economic factors, and vice-versa. The physical environment and health behavior domains were not associated with BAA–nHW disparities in AHM non-adherence.

### 3.2. Spatial Distribution in Heart Disease and Stroke Mortality

In Figure 2A, we show the posterior mean difference in the estimated log rate of heart disease mortality between BAAs and nHWs by county and the posterior probability that the log rate is higher in BAAs. The nHWs appeared to demonstrate higher rates of heart disease and stroke in some inland areas and in the northeast. Figure 2B shows that the posterior mean difference in the estimated log rate of stroke mortality between BAAs and nHWs by county and the posterior probability that the log rate was higher in BAAs. Strikingly, we observed higher rates of stroke mortality for BAAs across most of the counties studied. Figure 3 shows that the posterior mean estimate of the county race-specific factor and the posterior probability were above average (i.e., greater than 0), which implies the presence of unmeasured race-specific risk factors shared across cardiovascular outcomes. The factor estimate reflects unmeasured shared risk factors that impact both stroke and heart disease mortality; that is, risk unaccounted for by the other measured covariates included in the model, i.e., an estimate greater than zero would reflect an unmeasured increase in risk of mortality, and an estimate less than zero would reflect unmeasured risk of mortality. While there did not appear to be strong spatial patterns, we did observe areas with above-average factors across the south and west and below-average factors along the east coast for both races.

### 3.3. Associations between AHM Non-Adherence and Heart Disease and Stroke Mortality by Race/Ethnicity

AHM non-adherence was associated with both heart disease and stroke mortality among both BAAs and nHWs after adjusting for determinants of health and demographic factors (percentage BAA, percentage over 65 years old, percentage female, percentage rural), Table 3. For every standard deviation increase of 1 in the county-level prevalence of AHM non-adherence among BAAs, the county-level rates of both heart disease (β = 0.043; 95% CI: 0.027, 0.058) and stroke (β = 0.043; 95% CI: 0.023, 0.065) mortality increased among BAAs. For nHWs, for every standard deviation increase of 1 in the county-level prevalence of AHM non-adherence, county-level rates of heart disease and stroke mortality increased by 6.5% (β = 0.065; 95% CI: 0.047, 0.082) and 3.2% (β = 0.032; 95% CI: 0.011, 0.052), respectively. 

### 3.4. The Impact of Determinants of Health on the Effects of AHM Non-Adherence on Heart Disease and Stroke Mortality by Race/Ethnicity

Only the determinants of health in the clinical care and physical environment domains moderated the effects of AHM non-adherence on heart disease mortality among BAAs after adjusting for demographic factors (percentage BAA, percent over 65 years old, percent female, percent rural), Table 4. Among BAAs, the effects of AHM non-adherence on heart disease mortality decreased as the proportion of the population who had access to affordable, quality, and timely health care increased (β = −0.020; 95% CI: −0.038, −0.002). Similarly, as the proportion of the population who live in environments with high air and water quality and have access to quality housing and transportation increased, the effects of AHM non-adherence on heart disease mortality decreased, β = −0.018; 95% CI: −0.031, −0.004. The other two determinants of health factors (social and economic, and health behavior) did not moderate the effects of AHM non-adherence on heart disease mortality among BAAs. For stroke mortality, none of the health determinants were observed to moderate the effects of AHM non-adherence on stroke mortality among BAAs. For nHWs, none of the determinants of health moderated the effects of AHM non-adherence on heart disease mortality; only social and economic factors moderated (β = −0.027; 95% CI: −0.049, −0.005) the effects of AHM non-adherence on stroke mortality among nHWs. To further investigate why social and economic factors moderated the associations between AHM nonadherence and stroke mortality among nHWs but not among BAAs, we assessed the geographic variation in the association between social and economic factors and AHM non-adherence. We observed that social and economic factors were found to be associated with AHM non-adherence among both BAAs (β = 1.30; 95% CI: 0.24, 2.36) and nHWs (β = 2.83; 95% CI: 1.92, 3.74) only in the Northeast, the wealthiest region in the US, but not in the South, the poorest region; BAA (β = 0.43; 95% CI: −0.14, 1.01), nHWs (β = 0.33; 95% CI: −0.10, 0.76). 

## 4. Discussion

Our data have shown that the contextual determinants of health, especially clinical care and socioeconomic factors, are strongly associated with the BAA–nHW disparities in AHM non-adherence. For the associations between AHM non-adherence and deaths due to heart diseases, it appears that the contextual determinants of health (clinical care and physical environment) may moderate the adverse effects of AHM non-adherence on heart disease mortality among BAAs but not among nHWs. The contextual determinants of health did not moderate the associations between AHM non-adherence and stroke mortality among BAAs and only marginally moderated this association among nHWs through socioeconomic factors.

Our results confirm that AHM non-adherence is an independent predictor of heart disease and stroke mortality [4,13,14,15]. We extended this body of research by providing data that compared the impact of AHM non-adherence between BAAs and nHWs and found that AHM non-adherence is an independent predictor of heart disease and stroke mortality among both BAAs and nHWs. While the efficacy of AHM for controlling high blood pressure among both BAAs and nHWs is well established [41,42,43], these findings showed that AHM therapy could potentially become less effective when patients do not adhere to AHMs. Based on this and the fact that uncontrolled high blood pressure is a major risk factor for heart disease and stroke mortality, it stands to reason that AHM non-adherence can increase the risk of heart disease and stroke mortality through its adverse impact on high blood pressure control.

Our analysis has shown that the moderating effects of contextual determinants of health on the associations between AHM nonadherence and heart disease mortality differs by race/ethnicity. The role of the contextual determinants of health as contributors to the BAA–nHW disparities in AHM non-adherence and as potential moderators of the non-adherence-mortality association is a novel finding. Of the four domains of the contextual determinants of health, clinical care was the only factor that was both associated with BAA–nHW disparities in AHM non-adherence and moderated the AHM non-adherence-heart disease mortality relationship among BAAs. These data suggest that if clinical care, i.e., access to clinical care (defined as affordable, quality, and timely health care) were to be improved among BAAs, the BAA–nHW gaps in AHM non-adherence could be reduced. Consequently, improved adherence to AHM as a result of improved clinical care could potentially reduce the risk of heart disease deaths among BAAs and further reduce the BAA–nHW disparities in heart disease mortality.

The lack of association between the physical environment and AHM non-adherence could be an artifact of the heterogeneous structure of the physical environment construct, defined as a measure of quality of air and water, access to stable housing and quality transportation. Although some features of the physical environment such as housing instability and lack of quality transportation opportunities are known determinants of AHM non-adherence, there are no published data that suggest other features of the physical environment, air and water quality, are associated with AHM non-adherence. Air quality, access to stable housing and quality transportation were all inversely associated with heart disease mortality [44,45]. This led us to conclude that improving features of the built environment could potentially directly offset the adverse impact of AHM non-adherence on heart disease among BAA populations.

For stroke mortality, our findings suggest that determinants of health may not be involved in the pathway between AHM non-adherence and stroke mortality, especially among BAAs. While both clinical care and the social and economic domains of determinants of health were inversely associated with BAA–nHW AHM non-adherence gaps, only social and economic factors moderated the adverse impact of AHM non-adherence among nHWs, but not among BAAs. This was an unexpected finding given that low socioeconomic status, a known predictor of stroke mortality, is more prevalent among BAAs. Our exploratory analysis to assess potential geographic variation in the association between social and economic factors and AHM non-adherence may partly explain this unexpected finding. Our exploratory results suggest that the risk of stroke mortality may largely be a function of AHM non-adherence in low socioeconomic environments, whereas in more affluent environments, low socioeconomic status may moderate the adverse effects of AHM non-adherence on stroke mortality.

Adherence to medications is associated with adherence to healthy lifestyles [46,47]; it was therefore surprising that the health behavior construct was neither a predictor of BAA–nHW disparities in AHM non-adherence nor a moderator of the relationship between non-adherence and mortality due to heart disease and stroke among both BAAs and nHWs. Given that the health behavior construct was associated with a reduction in BAA–nHW disparities in AHM non-adherence in both a bivariate and an age-adjusted model (β = −0.17, *p* < 0.001) but not in fully-adjusted models (Table 2), it is possible that over-adjustment bias [48] could have partly explained why the health behavior construct was not a predictor of BAA–nHW disparities in AHM non-adherence. The inclusion of factors that are known determinants of healthy lifestyle choices—socioeconomic factors [49], the built environment [50,51], sex [52] and rural status [53]—could have resulted in an over-adjustment of the association between health behavior and BAA–nHW disparities in AHM non-adherence. With respect to the lack of evidence of health behavior as an effect moderator, it is possible that our finding was ecologically fallible, i.e., the findings from aggregate-level data may not translate to individual-level effects. In other words, if we were to replicate our analysis using individual-level health behavior and AMH non-adherence data, it is possible that health behavior would be an effect moderator given that health behaviors play a critical role in the etiology of high blood pressure control [54,55].

## 5. Limitations

Our analysis features some limitations that should be considered in the interpretation of our findings. First, because we investigated the associations between county-level AHM non-adherence, determinants of health and heart disease and stroke mortality, our findings may be subject to potential ecological fallacy [28,56]. Therefore, our findings cannot be used to infer the associations between individual-level measures of AHM non-adherence, determinants of health and heart disease and stroke mortality. Multilevel data that includes individual measures of AHM non-adherence and heart disease death and stroke mortality and information on patient–provider relationships [57], especially trust [58], a critical determinant of medication adherence especially among BAAs, are needed to confirm our findings. While adherence is an individualized behavior, population-level adherence is an important study outcome because: (1) small changes in population-level adherence could result in larger benefits in population health outcomes such as lower rates of hospitalizations and health care costs [59,60,61]; (2) population-level adherence measures are required for developing interventions geared at improving adherence among groups of patients [59,60]; (3) population-level adherence is increasingly being used as a quality indicator for the performance of health systems and individual physicians [62,63]. Second, PDC does not reflect primary non-adherence and does not account for gaps in medication refills during hospitalization and out-of-pocket payment for medications. Third, only a third of the counties featured sufficient counts of both BAA and nHW Medicare Part D beneficiaries who had measures of AHM non-adherence and heart disease and stroke mortality. Furthermore, the CDC Atlas dataset only included measures of AHM non-adherence among Medicare Part D beneficiaries. Thus, our findings may not be generalizable to older adults without Medicare prescription drug coverage or to younger adults. Fourth, although the CHR database is one of the most comprehensive databases on determinants of health, it contained limited factors about policy (social, health, and economic) and the built environment.

## 6. Strengths

In spite of the limitations discussed above, our analyses feature several strengths that could help to advance research on this topic. First, to the best of our knowledge, this is the first published research on the magnitude of the adverse impact of AHM non-adherence at the population level. Second, the application of rigorous state-of-art analytic techniques that employed a Bayesian paradigm enabled us to examine the role of the determinants of health as potential moderators of the relationships between AHM non-adherence and heart disease and stroke mortality among BAAs and nHWs. Third, this analysis revealed potential differential moderating effects of the determinants of health on adverse outcomes of AHM non-adherence by disease condition and race/ethnicity.

## 7. Conclusions

Our results confirm that AHM non-adherence is associated with an increased risk of both heart disease and stroke mortality among both BAAs and nHWs. Therefore, AHM non-adherence should be prioritized as a target for preventing heart disease and stroke mortality. Our findings suggest that to reduce BAA–-nHW disparities in AHM non-adherence, the clinical care and social and economic conditions in the counties in which BAA patients reside must be improved. Our findings further suggest that clinical care and features of the physical environment should be targeted in an effort to reduce the impact of AHM non-adherence on heart disease mortality among BAA patients. Implementing and evaluating interventions based on these findings could yield important insights into how to reduce the BAA–nHW disparities in heart disease mortality.

## Figures and Tables

**Figure 1 ijerph-18-12702-f001:**
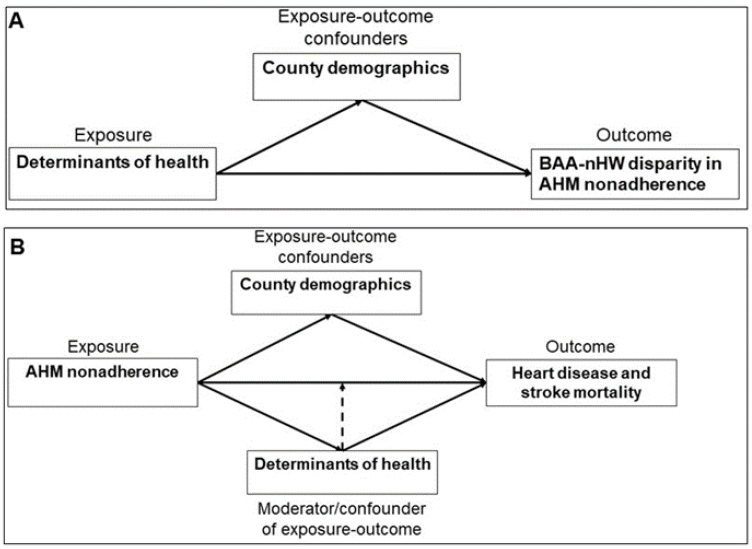
Conceptual models for accessing determinants of health as contributors of the BAA–nHW disparities in AHM non-adherence (**A**) and as potential moderators of the associations between AHM non-adherence and heart disease and stroke mortality (**B**). Solid arrows represent confounding effects while the dashed arrow represents moderating effects.

**Figure 2 ijerph-18-12702-f002:**
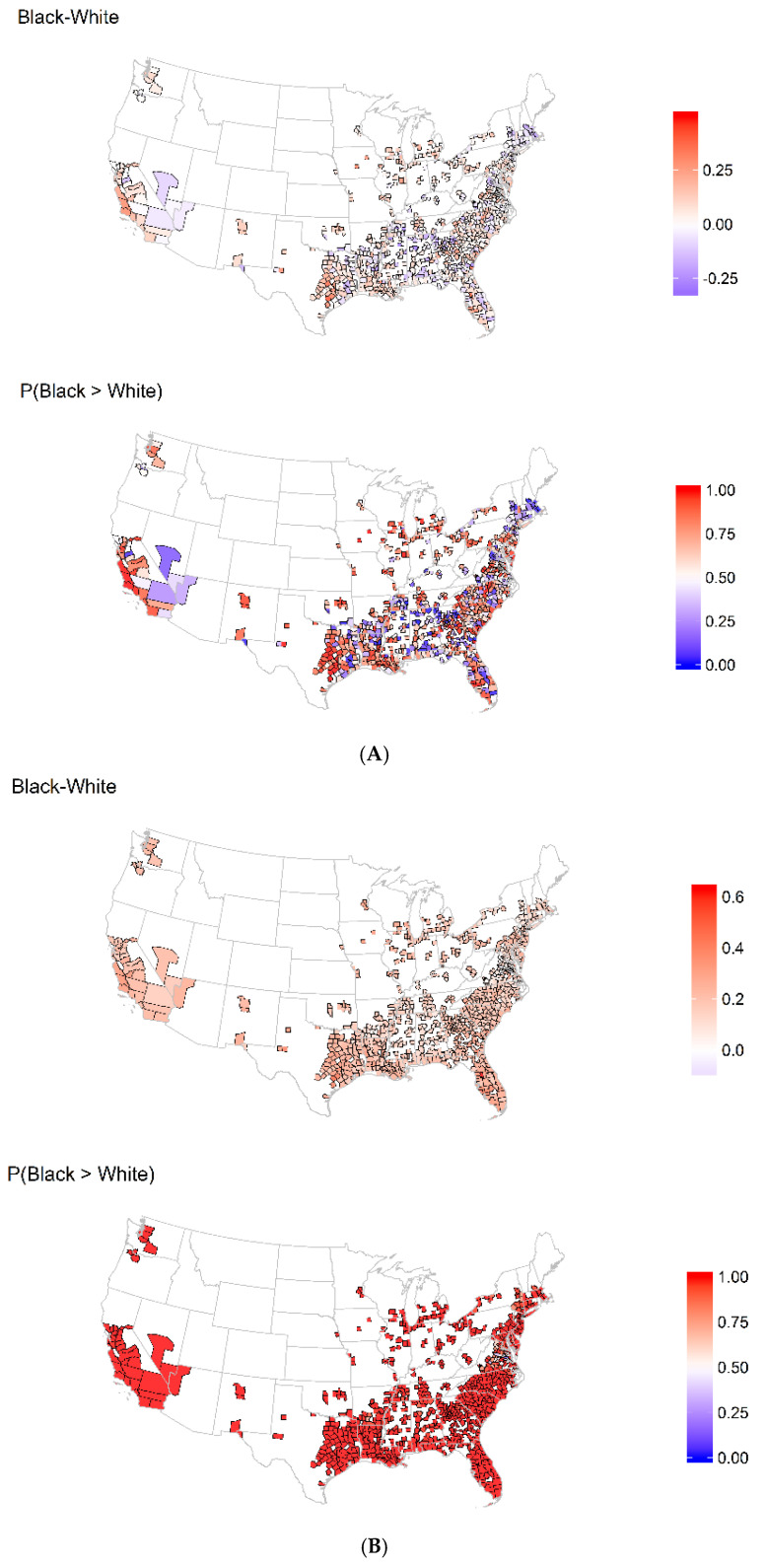
Spatial distribution of BAA–nHW disparities in log rate of heart diseases and stroke mortality among 875 counties in the US. (**A**) Heart Disease; (**B**) Stroke.

**Figure 3 ijerph-18-12702-f003:**
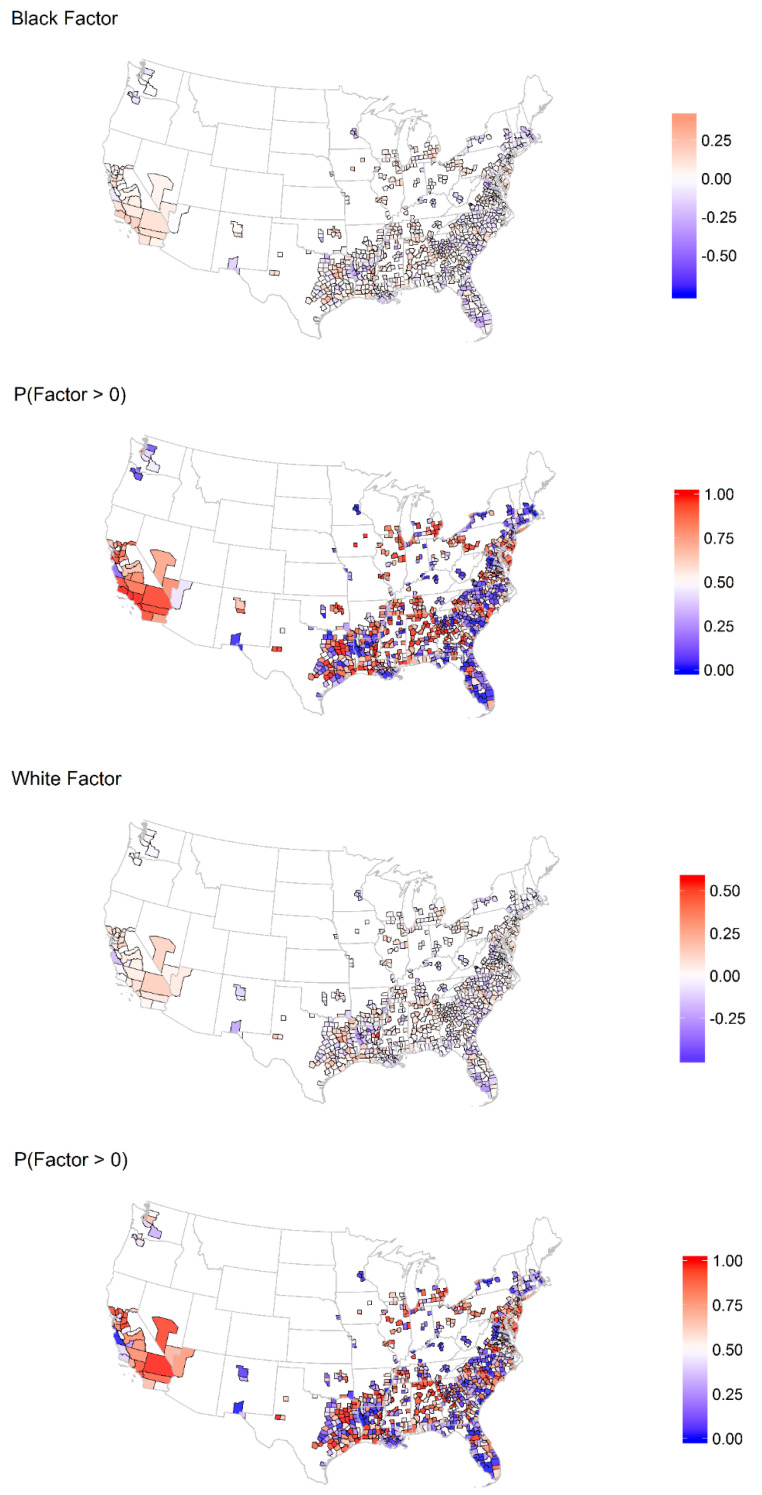
Spatial distribution of posterior mean estimate of the county race-specific factor and the posterior probability that it is above average.

**Table 1 ijerph-18-12702-t001:** Mean and standard deviations of variables included as covariates in the model by region among 875 counties in the U.S.

	Means Scores (Standard Deviation)				
Variable	Overall	Midwest	Northeast	South	West
**Measures of non-adherence**					
BAA AHM non-adherence (%)	34.6 (3.4)	32.1 (1.8)	31.9 (2.2)	35.5 (3.2)	31.8 (2.8)
White AHM non-adherence (%)	25.5 (3.4)	21.2 (2.0)	21.5 (2.0)	26.7 (2.8)	23.8 (2.3)
**Determinants of health scores**					
Health Behavior	0.06 (0.68)	−0.07 (0.48)	−0.65 (0.47)	0.23 (0.63)	−0.83 (0.50)
Clinical Care	0.02 (0.56)	−0.43 (0.37)	−0.53 (0.39)	0.17 (0.52)	−0.19 (0.40)
Physical Environment	0.18 (0.39)	0.27 (0.35)	0.09 (0.23)	0.21 (0.40)	−0.26 (0.36)
Social Economic Factors (%)	0.21 (0.69)	−0.17 (0.61)	−0.25 (0.57)	0.33 (0.67)	0.17 (0.68)
**Demographic factors**					
Black/African American population (%)	20.5 (16.0)	11.4 (8.5)	9.7 (7.8)	24.3 (16.2)	4.1 (3.2)
Over Age 65 (%)	15.9 (4.1)	14.7 (2.4)	16.1 (2.7)	16.2 (4.4)	13.8 (3.6)
Female (%)	50.7 (1.9)	51.0 (0.73)	51.2 (0.75)	50.7 (2.2)	50.1 (1.1)
Rural (%)	39.5 (29.1)	17.7 (15.3)	17.2 (18.0)	47.5 (28.3)	9.9 (8.4)

Negative mean scores signify that the average scores of a factor is below the average from the overall counties included in the original analysis.

**Table 2 ijerph-18-12702-t002:** Associations between constructs of determinants of health and the gap in non-adherence to antihypertensive medications between Black/African Americans and non-Hispanic Whites.

Constructs of Determinants of Health	Regression Coefficients			
	Model 1	Model 2	Model 3	Model 4
Health Behaviors	−0.17 **	−0.17 **	0.05	0.11
Clinical Care	−0.31 **	−0.31 **	−0.30 **	−0.21 **
Social and Economic	−0.16 **	−0.16 **	−0.06	−0.11 *
Physical Environment	−0.03	−0.04	−0.01	0.00
Pseudo-R^2^	N/A	N/A	0.24	0.25

Model 1: Bivariate models, including individual constructs of determinants of health; Model 2: age-adjusted models; Model 3: Includes all four constructs of determinants of health; Model 4: multivariate associations between constructs of determinants of health adjusted by county-level percent BAA population, percent female population, percent ≥65 years old and percent rural area * *p* < 0.05; ** *p* < 0.001.

**Table 3 ijerph-18-12702-t003:** Estimated posterior mean change of the average log rate of heart disease and stroke mortality for Black/African Americans and non-Hispanic Whites for a standard deviation increase of 1 in each covariate conditional on mean values of all other covariates among 875 counties in the U.S.

Predictor	BAA Heart Disease		BAA Stroke		nHW Heart Disease		nHW Stroke	
	Posterior Mean Estimate (95% CI)	*p* (Effect > 0)	Posterior Mean Estimate (95% CI)	*p* (Effect > 0)	Posterior Mean Estimate (95% CI)	*p* (Effect > 0)	Posterior Mean Estimate (95% CI)	*p* (Effect > 0)
BAA AHM Non-adherence	0.043 (0.027, 0.058)	1.00	0.043 (0.023, 0.065)	1.00	n/a	n/a	n/a	n/a
nHW AHM Non-adherence	n/a	n/a	n/a		0.065 (0.047, 0.082)	1.00	0.032 (0.011, 0.052)	1.00

All effects are adjusted for county-level percentage BAA population, percentage female population, percentage ≥65 years old and percentage rural area. Abbreviations: BAA, Black/African American; nHW, non-Hispanic White; AHM, antihypertensive medication.

**Table 4 ijerph-18-12702-t004:** Estimated posterior mean change in the effect of AHM non-adherence on the average log rate of heart disease and stroke for Black/African Americans and non-Hispanic Whites for a standard deviation increase of 1 in each covariate among 875 counties in the U.S.

	BAA Heart Disease		BAA Stroke		White Heart Disease		White Stroke	
	Posterior Mean Estimate (95% CI)	*p* (Effect > 0)	Posterior Mean Estimate (95% CI)	*p* (Effect > 0)	Posterior Mean Estimate (95% CI)	*p* (Effect > 0)	Posterior Mean Estimate (95% CI)	*p* (Effect > 0)
AHM non-adherence interaction with:								
Health Behavior	−0.011 (−0.031, 0.009)	0.14	−0.010 (−0.035, 0.015)	0.21	−0.009 (−0.028, 0.009)	0.16	0.015 (−0.006, 0.037)	0.92
Clinical Care	−0.020 (−0.038, −0.002)	0.02	0.006 (−0.018, 0.031)	0.71	0.004 (−0.012, 0.020)	0.68	−0.008 (−0.025, 0.011)	0.20
Physical Environment	−0.018 (−0.031, −0.004)	0.01	−0.010 (−0.028, 0.007)	0.14	−0.007 (−0.019, 0.005)	0.13	−0.009 (−0.023, 0.005)	0.11
Social and economic Factors	0.014 (−0.008, 0.035)	0.89	−0.018 (−0.046, 0.008)	0.09	0.007 (−0.013, 0.026)	0.76	−0.027 (−0.049, −0.005)	0.01

All effects are adjusted for
county-level percent BAA population, percentage female population, percentage ≥65 years old and percentage rural area. Abbreviations: BAA, Black/African American; nHW, non-Hispanic White; AHM, antihypertensive medication.

## Data Availability

Not applicable.

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
