# Peer review of "Examining the Drivers of Racial/Ethnic Disparities in Non-Adherence to Antihypertensive Medications and Mortality Due to Heart Disease and Stroke: A County-Level Analysis"

_ijerph, 2021, doi:10.3390/ijerph182312702_

Round 1

Reviewer 1 Report

Thank you for the interesting paper on racial/ethnic disparities in medication non-adherence in the US. I really liked the focus on the (social) determinants of health in this context. The vast majority of available literature on health inqualities relies on individual-level data, that does not (necessarily) cover the characteristics of wider social and physical environments where these people live. However, your approach also poses a risk for ecological fallacy (as you also list under Limitations). Following this line of argument, I had a few questions/comments:

a) Lines 243-246:  The physical environment and health behavior domains were not associated with BAA-nHW disparities in AHM non-adherence - Given the available evidence on the role of behavioural factors in heart disease etiology and also to hypertension control (e.g. 10.1038/s41598-018-26823-5)

could you please a brief explanation to your findings and the potential of ecologic fallacy (as individual lifestyle factors can not be accounted for).

b) Could the higher importance of clinical care (compared to behavioural and SES factors) in AHM adherence be at least partly be explained by its smaller variance (healthcare setting being more homogenous than behavioural and socio-economic factors)?

c) Lines 399-400: Did you also consider pooling the county level data and analyzing it in clusters defined by either share of BAA and/or wealth (mean income or other relevant measure)?

Other than these discussion points, I believe that this paper is generally well written and has definately a merit for readers of IJERPH.

Reviewer 2 Report

This manuscript has several strengths that are well described in the "Strengths" section. The limitations are also well described from a largely statistical perspective, but other limitations exist. Namely, the co-authors have failed to explicitly consider dis/mistrust of science by BAA, and racism/discrimination as factors in AHM non-adherence. 

Co-authors are encouraged to cite the work of Corbie-Smith, and others, to document dis/mistrust, and to acknowledge that this factor was not/cannot be analyzed with the methods described.

With respect to racism/discrimination there is the possibility that the co-authors can examine greater exposure to this factor as a measure of residential segregation. This metric is explored in Warner and Gomez, 2010 (J Community Health. 2010 August ; 35(4): 398–408). In this paper extreme segregation is associated with better breast cancer outcomes for BAA. Therefore, it is possible that this better health outcome could be attributed to lesser racism/discrimination, greater social support, and access to more BAA healthcare professionals. Greater access to BAA healthcare professionals can overcome dis/mistrust and improve clinical care.
